# Shorter Door-to-ECG Time Is Associated with Improved Mortality in STEMI Patients

**DOI:** 10.3390/jcm13092650

**Published:** 2024-04-30

**Authors:** Maame Yaa A. B. Yiadom, Wu Gong, Sean M. Bloos, Gabrielle Bunney, Rana Kabeer, Melissa A. Pasao, Fatima Rodriguez, Christopher W. Baugh, Angela M. Mills, Nicholas Gavin, Seth R. Podolsky, Gilberto A. Salazar, Brian Patterson, Bryn E. Mumma, Mary E. Tanski, Dandan Liu

**Affiliations:** 1Department of Emergency Medicine, Stanford University, 900 Welch Road, Ste 350, Palo Alto, CA 94304, USA; gbunney@stanford.edu (G.B.); kabeer@stanford.edu (R.K.); mapasao@stanford.edu (M.A.P.); 2Department of Biostatistics, Vanderbilt University Medical Center, 2525 West End Avenue, Suite 1100, Nashville, TN 37203, USA; wu.gong@outlook.com (W.G.); dandan.liu@vumc.org (D.L.); 3Tulane University School of Medicine, 1430 Tulane Ave., New Orleans, LA 70112, USA; sbloos@stanford.edu; 4Division of Cardiovascular Medicine, Department of Medicine, Stanford University, 6453 Quarry Rd., Palo Alto, CA 94304, USA; frodrigu@stanford.edu; 5Department of Emergency Medicine, Brigham and Women’s Hospital, Harvard University, 75 Francis Street, Boston, MA 02115, USA; cbaugh@bwh.harvard.edu; 6Department of Emergency Medicine, Columbia University College of Physicians and Surgeons, 622 W 168th St., New York, NY 10032, USA; amm2513@cumc.columbia.edu; 7Department of Emergency Medicine, Icahn School of Medicine at Mount Sinai, 1468 Madison Ave., New York, NY 10029, USA; nicholas.gavin@mountsinai.org; 8Department of Dean Administration, Oregon Health & Sciences University, 3181 SW Sam Jackson Park Rd., Portland, OR 97239, USA; podolsky@ohsu.edu; 9Department of Emergency Medicine Parkland Hospital, University of Texas Southwestern Medical Center—Dallas, 5323 Harry Hines Boulevard E4.300, Dallas, TX 75390, USA; gilberto.salazar@utsouthwestern.edu; 10Department of Emergency Medicine, University of Wisconsin—Madison, 800 University Bay Dr Suite 310, Madison, WI 53705, USA; bpatter@medicine.wisc.edu; 11Department of Emergency Medicine, University of California at Davis, 2245 45th St., Sacramento, CA 95817, USA; 12Department of Emergency Medicine, Oregon Health & Science University, 3181 SW Sam Jackson Park Rd., Portland, OR 97239, USA; tanski@ohsu.edu

**Keywords:** door-to-ECG, electrocardiogram, STEMI, percutaneous coronary intervention

## Abstract

**Background:** Delayed intervention for ST-segment elevation myocardial infarction (STEMI) is associated with higher mortality. The association of door-to-ECG (D2E) with clinical outcomes has not been directly explored in a contemporary US-based population. **Methods:** This was a three-year, 10-center, retrospective cohort study of ED-diagnosed patients with STEMI comparing mortality between those who received timely (<10 min) vs. untimely (>10 min) diagnostic ECG. Among survivors, we explored left ventricular ejection fraction (LVEF) dysfunction during the STEMI encounter and recovery upon post-discharge follow-up. **Results:** Mortality was lower among those who received a timely ECG where one-week mortality was 5% (21/420) vs. 10.2% (26/256) among those with untimely ECGs (*p* = 0.016), and in-hospital mortality was 6.0% (25/420) vs. 10.9% (28/256) (*p* = 0.028). Data to compare change in LVEF metrics were available in only 24% of patients during the STEMI encounter and 46.5% on discharge follow-up. **Conclusions:** D2E within 10 min may be associated with a 50% reduction in mortality among ED STEMI patients. LVEF dysfunction is the primary resultant morbidity among STEMI survivors but was infrequently assessed despite low LVEF being an indication for survival-improving therapy. It will be difficult to assess the impact of STEMI care interventions without more consistent LVEF assessment.

## 1. Introduction

ST-elevation myocardial infarction (STEMI) occurs due to the occlusion of one or more coronary arteries, resulting in transmural myocardial ischemia [1]. Without prompt diagnosis, STEMI is associated with high mortality and increased heart failure incidence [1,2,3,4,5,6,7]. Specifically, after treating STEMI with the preferred therapy of percutaneous coronary intervention (PCI), in-hospital mortality has been reported to be reduced from 30–40% to approximately 6–7% [3,4,5,6,7]. Early intervention is also associated with less heart failure from reduced ischemia time in survivors [8,9,10]. Left ventricular ejection fraction (LVEF) in STEMI patients is an important measure used to identify patients with heart failure with reduced ejection fractions (HFrEF). Those with LVEF below 50% have worse prognoses [10,11,12,13].

Achieving best STEMI outcomes for those receiving PCI requires timely care along the continuum from prompt diagnosis to coronary revascularization [14,15]. As a result, clinical practice guidelines from the American Heart Association (AHA), American College of Cardiology (ACC), and the European Society of Cardiology recommend prompt recognition and diagnosis of STEMI through regional STEMI care processes [2,13,14]. This begins with early diagnosis upon first medical contact [16,17]. Our prior work has shown that STEMI patients with a delayed acquisition of a diagnostic electrocardiogram (ECG) have a longer door-to-balloon (D2B) time [18,19,20]. We have also observed how the individuals and care teams making the diagnosis vary depending on the location of diagnosis with emergency department (ED) diagnosed patients being the least studied [21].

The associations between diagnostic delay and subsequent mortality and morbidity (e.g., heart failure) have not been directly quantified. Here, we explored a subgroup of ED-diagnosed STEMI patients [21] to better understand whether ED-based interventions to reduce door-to-electrocardiogram (D2E) time may improve mortality and reduce new onset heart failure in survivors. In this descriptive investigation, we compare the mortality rate among STEMI patients who did and those who did not receive a timely ECG, while exploring differences in LVEF. We hypothesized that STEMI patients with timely ECG would have improved clinical outcomes as a result of more timely treatment. To our knowledge this is the only exploration of this kind within a contemporary US-based population.

## 2. Methods

We used data included in the Emergency STEMI Care Registry (ESC) from 10 PCI-center associated EDs [18,19,20,21]. The 10 EDs included Brigham and Women’s Hospital in Boston, Massachusetts; NYU—Langone and New York-Presbyterian Columbia University in New York, New York; University of Pennsylvania in Philadelphia, Pennsylvania; Vanderbilt University in Nashville, Tennessee; University of Wisconsin in Madison, Wisconsin; The Cleveland Clinic Foundation Main Campus in Cleveland, Ohio; University of Texas Southwestern affiliated Parkland Hospital in Dallas, Texas; Oregon Health & Science University in Portland, Oregon; and University of California, Davis Medical Center in Sacramento, California.

These facilities contributed patient data for all patients with STEMI seen from 1 January 2014–31 December 2016 to the ESC Registry, described below and in prior publications [18,19,20,21,22]. Our description here partly reproduces their wording. The diagnosis of STEMI was based on final hospital international classification of disease (ICD) diagnosis codes. To capture the ED-diagnosed cohort [21], we excluded patients with a STEMI diagnosis or screening ECG completed prior to ED arrival, a non-diagnostic initial ECG, and in-hospital STEMI. In-hospital STEMI was defined as instances when a diagnostic ECG was acquired after hospital admission for an alternative diagnosis and when an ED ECG was present and without evidence of STEMI. We then excluded patients whose catheterization lab findings were not consistent with STEMI (e.g., no culprit/coronary artery lesion found at time of invasive angiography) and when there was an alternative diagnosis for which care was more consistent.

### 2.1. Comparison Cohorts and Outcomes

Our analyses compared differences between those achieving timely diagnosis and those who did not. Timely diagnosis was defined as a D2E ≤ 10 min per international STEMI management guidelines, and untimely diagnosis was a D2E > 10 min [1,2,18,19,20,21].

### 2.2. Outcomes

Our primary focus was one-week and in-hospital mortality, which was compared between those who did and did not receive a timely ECG in adherence with international diagnosis guidelines [2,13,14]. We also examined these outcomes in the subgroup of ED STEMI patients who received PCI.

When examining LVEF, we considered that the longer a STEMI persists before intervention, the greater the loss of viable myocardial tissue. During recovery, the myocardium remodels to heal [15,23]. This remodeling can lead to ventricular chamber scarring and dilation that often reduces the proportion of blood volume ejected with each cardiac contraction [8,10]. As a result, we conducted an exploratory analysis evaluating three ventricular function metrics, each of which quantified a change in LVEF (Figure 1). Consistent with guidelines for STEMI care measures, the ESC registry data collection included LVEF assessment made using echocardiography, catheterization laboratory coronary angiography, or coronary CT angiography [3,4,14]. ***STEMI LVEF dysfunction*** was defined as the difference in lowest LVEF assessment documented prior to the STEMI compared to the lowest assessed during the STEMI. This represents a reduction in pump function while the myocardium is ischemic. ***Post-STEMI LVEF dysfunction*** was defined as the difference in LVEF documented prior to the STEMI compared to the first documented after hospital discharge upon follow-up. ***Post-STEMI LVEF recovery*** was defined as the difference in the lowest LVEF assessed during the STEMI episode while the myocardium is ischemic to the first documented after hospital discharge upon follow-up.

### 2.3. Data Collection

The ESC Registry is an emergency care delivery-focused data registry [19,20,21,22]. We acquired the cohort of potentially eligible patients using electronic health record (EHR) data abstraction of ED STEMI patients seen during the study period using ICD-coded final hospital diagnoses consistent with acute STEMI. Individual patient care details were obtained with manual chart review by data abstractors. Each abstractor received two hours of standardized training including a 90-min training module with practice data collection and data accuracy verification by the data coordinating center. The process and details of their training and multi-centered data collection have been previously published [18]. Cases were screened and flagged for potential exclusion during chart review using the pre-specified criteria noted above. The study principal investigator and specific site-principal investigators reviewed flagged cases to verify exclusion.

All study data were maintained in a customized REDCap database (REDCap; https://www.project-redcap.org, accessed on 18 March 2023) [24]. The protocol for data collection was approved by the Institutional Review Board at each site. Informed consent was waived due to the retrospective nature of the study.

### 2.4. Statistical Analysis

Group comparisons were conducted using Wilcoxon rank sum test for continuous variables and Chi-square test for categorical variables. Fisher’s exact test was used for categorical variables with any cell count less than 5. We used a complete case cohort for each outcome variable and reported available data in all tables, using an 0.05 alpha-level of significance for all comparisons without adjustments. All analyses were performed using the R statistical software, Version 3.4.2.

### 2.5. Data Availability Statement

Study data can be shared upon request with scientific review by the Emergency STEMI Care Registry and ethics review by the ESC’s affiliated institution and that of the requestor and establishment of a data use agreement for sharing.

## 3. Results

There were 676 ED-diagnosed patients across the 10 sites who experienced STEMI, of which 85.2% (576) received PCI. As previously reported for this ED-diagnosed cohort, 62.1% (420) received a timely ECG [19,23]. Among those who received PCI, 65.8% (379) received a timely ECG. Further, the median D2E among patients who received a timely ECG was 5 min (IQR: 3.7); the value for those who received an untimely ECG was 18 min (IQR: 14.40) (*p* < 0.001). Demographic differences between those receiving timely ECG vs. untimely ECG have been previously reported [19,23]. We saw no significant differences in baseline comorbidities or hospital length of stay (LOS) (Table 1 and Table 2).

### 3.1. Mortality

Among all ED-diagnosed patients, overall one-week mortality was 7.0% (47/676), and in-hospital mortality was 7.8% (53/676). Deaths were fewer among those who received a timely ECG, where one-week mortality was 5% (21/420) versus 10.2% (26/256) among those with untimely ECGs (*p* = 0.016), and in-hospital mortality was 6.0% (25/420) versus 10.9% (28/256) (*p* = 0.028). This supports the practice assumption that an earlier ECG enables earlier treatment, resulting in lower mortality (Table 2). Further, we observed that 43% (20/47) of deaths within one week and 39.6% (21/53) of those occurring in-hospital were among the 14.7% (100/676) who did not receive PCI. In this group, one-week and in-hospital mortality were 20% (20/100) and 21% (21/100), respectively (Table 2).

PCI within 90 min of initial hospital presentation has been associated with up to a 50% reduction in mortality among cardiac arrest STEMI patients and general STEMI patients in Europe and Australia [3,25,26]. We therefore examined mortality in the 576 patients who received PCI in our US-based cohort. We observed similar trends in our study, including a reduction from 6.6% (13/197) to 3.7% (14/379) for 1-week mortality and a reduction from 7.6% (15/197) to 4.5% (17/379) for in-hospital mortality; however, the *p*-values (0.175 and 0.173, respectively) are above the significance level of 0.05 (Table 3).

### 3.2. STEMI Left Ventricular Ejection Fraction Dysfunction and Recovery

#### 3.2.1. Dysfunction during the Acute STEMI Episode

We observed notably greater LVEF dysfunction during the STEMI encounter among those with untimely ECG at 10% vs. 7.5% (*p* = 0.057) for those with a timely diagnosis (Table 2). This finding was not statistically significant, but a notable magnitude difference was noted. We observed that there were more patients who did not receive PCI among those with an untimely ECG. In the subset of patients receiving PCI, there was no statistically significant difference in LVEF dysfunction between those with a timely or untimely ECG (6.5% vs. 9%, *p* = 0.408) (Table 4).

#### 3.2.2. Post-STEMI LVEF Recovery

We saw no difference in post-STEMI LVEF recovery between those who did and did not receive timely ECG in the dataset overall (Table 2). However, we observed an improvement among those receiving PCI who had a timely ECG compared to those who did not (5% vs. 0%, *p* = 0.024, Table 3).

#### 3.2.3. LVEF Assessment Compliance

With heart failure being the greatest morbidity factor after a STEMI, we expected LVEF dysfunction associated with the acute STEMI to be routinely documented in the hospital. However, we found that 25.1% of all patients had an LVEF assessment prior to their STEMI, 24.7% had an assessment during their STEMI hospitalization, and 45.5% had an assessment after discharge. Similarly, amongst patients who had a STEMI and underwent PCI treatment, 24% had an echo prior to their STEMI, 24.5% had an echo during their hospitalization, and 46.5% had an echo after discharge (Table 5). The rates of LVEF ascertainment were evenly distributed among groups, so there was no one characteristic that skewed the low amount of LVEF documentation.

## 4. Discussion

In our multi-centered geographically diverse, US cohort of ED-diagnosed STEMI patients, we observed over 50% lower in-hospital and 1-week mortality among those who received timely diagnostic testing compared to those who did not. Among those who received PCI we similarly observed an almost 50% reduction in mortality. Analogous observations among patients receiving PCI did not achieve statistical significance. However, our mortality findings are consistent, albeit slightly higher than, the one-week mortality of 3.4% observed in an Australian national cohort^3^ and the in-hospital mortality of 4.6% in a comparable US-based cohort including all STEMI patients (Table 3) [27,28]. This suggests potential clinical significance worthy of further investigation.

The limited LVEF that we observed in real-world documentation during STEMI hospitalization is concerning because prior work has noted LVEF assessment is associated with the initiation of evidence-based goal-directed medical therapy (GDMT) medications prior to discharge. Specifically, the early introduction of GDMT to these patients, which involves the initiation and titration of pharmacological agents, including angiotensin converting enzyme-inhibitors, angiotensin II receptor blockers, angiotensin receptor neprilysin inhibitors, beta-blockers, mineralocorticoid receptor antagonists, and sodium-glucose co-transporter-2 inhibitors, has been shown to reduce mortality and heart failure related hospitalizations [10,11,12]. Thus, determination of LVEF prior to discharge provides the opportunity to initiate GDMT in a timely manner and help to improve 30-day and one-year survival for patients with HFrEF [27,28,29,30,31].

Consequently, the paucity of LVEF documentation is very concerning considering that the 2017 AHA/ACC STEMI Care Quality Measures include an evaluation of LVEF [10]. This is not new guidance as the 2013 ACCF/AHA Guidelines for the Management of Patients with STEMI recommended the same [2]. There are three reasons why improving LVEF assessment compliance as a metric is important. First, LVEF < 50% may benefit from targeted medical therapies as noted above. Second, LVEF is one of the strongest predictors of long-term survival following acute myocardial infarction [28]. Lastly, it provides a baseline for reassessment after the acute episode to guide the potential need for device therapy, including left ventricular assist devices.

All the LVEF modalities included in our assessment are considered acceptable per guidelines with the most common, trans-thoracic echocardiography, being inexpensive and non-invasive. However, our findings suggest there is a practice barrier for obtaining LVEF assessment. This represents an opportunity for improvement that may potentially impact survival in HFrEF patients post STEMI. Compliance monitoring may close the gap and enhance patient outcomes.

STEMI mortality is highly associated with the quality of care received and events occurring within the first 30 days [8,9]. Earlier PCI also improves survival and LVEF [10,11,26]. The initiation of PCI is dependent on a timely diagnosis during the emergency care phase, and time to PCI has been found to be the strongest predictor of clinical outcomes, more so than risk factors and other patient characteristics, across multiple STEMI patient cohorts. Our previous studies emphasized that the time to the diagnostic ECG influences timely access to PCI and is a modifiable factor in care delivery [16,21]. Thus, efforts towards earlier diagnostic ECG for all STEMI patients may improve survival to hospital discharge. However, there are differentiating factors that should be considered when interpreting the results presented here related to our focus on the screening and diagnostic phase of care.

First, our patient cohort includes those diagnosed with the first ECG performed at a PCI-center ED. This inclusion criterion does not consider the 15% of ED STEMI patients who are not diagnosed with the first performed ED ECGs [17] or those who are diagnosed in an ambulance or in a referring ED. Despite these exclusions, the study cohort is likely to be representative of, and generalizable, to the ED STEMI population as whole. Although subsequent care may vary, diagnostic processes associated with D2E at these locations have marked similarities to all EDs across the US [2,3,4,14,20]. In addition, patients with prehospital diagnosis of STEMI and activation of the cardiac catheterization laboratory team have lower mortality [31].

Second, we use a pragmatic assessment of mortality and LVEF by using the information documented in each patient’s EHR. Our assessment of one-week and in-hospital mortality and presence of LVEF documentation was followed out to one year to permit potential lags in documentation. All sites used the same EHR vendor that includes a health-information exchange capability providing access to records of over 50% of hospitals in the US [32]. In addition, all data abstractors received training to ensure standard data collection. This increases our confidence that we collected outcome data as they were available to practicing providers. One possible limitation is that LVEF assessed informally using point of care ultrasound (POCUS) would not have been captured. However, POCUS is not recognized by the AHA as a modality meeting their metrics for LVEF measurement. Thus, our pragmatic data collection more closely resembles the data visible to contemporary providers involved in the in-hospital and post-hospital discharge care for STEMI patients.

Third, assessment of LVEF in our ED-diagnosed cohort was lower than that reported in a major ACTION Registry-Get with the Guidelines (GWTG) STEMI study from 10 years prior. That study included 50,863 patients from 379 hospital facilities and reported >90% in-hospital LVEF assessment compliance [26], which is markedly different than our observed value of 24.7%. This difference may be due to the ACTION-GWTG registry (1) excluding many STEMI patients due to “non-system delays” that are included in our cohort, (2) applying exclusions based on care outcomes despite this not affecting care delivery for screening and diagnosis, and (3) reflecting data collected during a period (2007–2009) when EHR systems were not routinely used. These factors likely contributed to the high variability in exclusion rate among ACTION-GWTG registry reporting facilities, which has been reported to be as high as 68% [33]. Many of those excluded were patients at risk for low LVEF, the very patients who may benefit from remodeling supportive therapy, and an additional 12.4% of patients were excluded after a STEMI diagnosis was made based on distal care events, including missing LVEF assessment. The ACTION-GWTG cohort’s orientation to study typical system activity by excluding these patients reduces generalizability to the more complete population of STEMI patients and increases the likelihood of over reporting LVEF assessment. Thus, the low LVEF assessment level in our cohort may represent a real-world practice pattern that would benefit from additional study.

Lastly, we looked for LVEF assessment measures before, during, and after hospitalization in order to determine change in LVEF. Since our exploratory analysis involved the comparison of two LVEF measurements, missing data for one of the measurements left us unable to calculate an outcome in a large proportion of patients. We present our findings here as a report of what was observed from a pragmatic clinical practice lens, with a more inclusive population of STEMI patients, diagnosed in a common care environment that offers focused intervention opportunities. Our findings raise concerns that the clinical impact of STEMI care delivery improvement efforts will be difficult to assess without more consistent LVEF assessment as part of acute and recovery STEMI care.

## 5. Conclusions

Achieving D2E within 10 min may be associated with a 50% reduction in mortality among ED STEMI patients. These findings further suggest that early diagnosis and early PCI have a significant impact on survival to hospital discharge and are consistent with observations seen in non-US based cohorts.

Heart failure is the primary resultant morbidity among STEMI survivors, often evidenced by a decrease in LVEF. We observed low rates of LVEF measurements in our nationally representative US-based cohort, despite international guidelines highlighting the prognostic importance of LVEF. It is difficult to assess the impact of acute and recovery STEMI care interventions without more consistent LVEF assessment and documentation. With LVEF measures guiding indications for GDMT therapy, increased LVEF documentation may improve survival and functional outcomes among STEMI patients with resultant HFrEF.

## Figures and Tables

**Figure 1 jcm-13-02650-f001:**
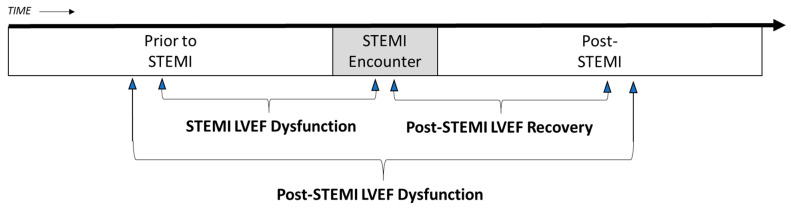
STEMI Associated LVEF Dysfunction and Recovery Metrics. Left ventricular ejection fraction (LVEF) disfunction before and after time intervals of interest as a metric for heart failure. *STEMI LVEF Dysfunction* is the lowest LVEF assessment during the emergency department to hospital stay associated with the acute STEMI encounter subtracted from the lowest assessment in the patient’s electronic health record prior to the STEMI encounter. *Post-STEMI LVEF Recovery* is the lowest LVEF assessment documented during the acute STEMI encounter subtracted from the lowest assessment documented within 1 year after hospital discharge. *Post-STEMI LVEF Dysfunction* is the lowest LVEF assessment documented within one year after hospital discharge subtracted from the lowest LVEF assessment in the patient’s electronic health record prior to the STEMI encounter.

**Table 1 jcm-13-02650-t001:** ED-Diagnosed STEMI Patients Receiving Timely (<10 min) vs. Untimely ECG (≥10 min) from the ESC Registry.

	All Patients *n* = 676	Timely Door-to-ECG (≤10 min) *n* = 420	Untimely Door-to-ECG (>10 min) *n* = 256	*p*-Value
Age (years) *	53 61 69	53 61 68	53 61 71	0.6176
Door-to-ECG (D2E) *	4.0 7.0 16.0	3.0 5.0 7.0	14.0 20.5 44.2	<0.001
Gender (Female)	26.5% (179)	22.6% (95)	32.8% (84)	0.005
Race				0.005
White	63% (427)	65.7% (276)	59% (151)	
Black or African American	16.6% (112)	12.4% (52)	23.4% (60)	
Non-white Latino	1.3% (9)	1.2% (5)	1.4% (4)	
Asian or Native American	6.5% (44)	6.9% (29)	5.9% (15)	
Unknown	12.4% (84)	13.8% (58)	10.2% (26)	
Ethnicity				0.027
Non-Hispanic	75.3% (509)	76.7% (322)	73% (187)	
Hispanic	14.2% (96)	15.2% (64)	12.5% (32)	
Unknown	10.5% (71)	8.1% (34)	14.5% (37)	
Primary Language				0.090
English	78.6% (531)	80.5% (338)	75.4% (193)	
Spanish	11.7% (79)	11.9% (50)	11.3% (29)	
Arabic	0.7% (5)	0.5% (2)	1.2% (3)	
Other	4.3% (29)	4.0% (17)	4.7% (12)	
Comorbidities				
Hypertension	66.6% (450)	65.2% (274)	68.8% (176)	0.393
Diabetes	34% (230)	30.2% (127)	40.2% (103)	0.010
Hyperlipidemia	56.2% (380)	59% (248)	51.6% (132)	0.068
Heart Failure	10.2% (69)	9.3% (39)	11.7% (30)	0.377
Prior MI	21.2% (143)	21.9% (92)	19.9% (51)	0.606
Prior PCI	19.1% (129)	19.5% (82)	18.4% (47)	0.785
Prior CABG	4% (27)	3.1% (13)	5.5% (14)	0.185
Smoking	24.3% (164)	22.1% (93)	27.7% (71)	0.121

Abbreviations: CABG = coronary artery bypass graft; ECG = electrocardiogram; ED = emergency department; ESC = Emergency STEMI Care Registry; MI = myocardial infarction; PCI = percutaneous coronary intervention. * a b c where a represents the lower quartile, b the median, and c the upper quartile for continuous variables. Numbers after proportions are frequencies.

**Table 2 jcm-13-02650-t002:** Outcomes for All ED-Diagnosed STEMI Patients Regardless of Treatment Modality.

	Available Data *n* = 676	Overall	Timely ECG *n* = 420 (62.1%)	Untimely ECG *n* = 256 (37.9%)	*p*-Value
Hospital LOS in days (median [IQR])	100% (676)	4 [3–6]	4 [3–6]	4 [3–6]	0.481
STEMI LVEF Dysfunction	19.5% (132)	−10% [−25 to 0%]	−7.5% [−20 to 0%]	−10% [−25.5 to −4.5%]	0.057
Post STEMI LVEF Dysfunction	14.2% (96)	−5% [−16.2 to 0%]	−5% [−15 to 0]	−0.5 [−9.5 to −5]	0.121
Post STEMI LVEF Recovery	14.6% (99)	5 [−10 to 1%]	5 [−10 to 0%]	5 [−9.5 to 5.0]	0.153
One-week Mortality	100% (676)	7% (47)	5% (21)	10.2% (26)	0.016 *
In-hospital Mortality	100% (676)	7.8% (53)	6.0% (25)	10.9% (28)	0.028 *
Received PCI	100% (676)	85% (576)	65% (379)	34% (197)	<0.001 *

Abbreviations: ECG = electrocardiogram; LOS = length of stay; LVEF = Left ventricular ejection fraction. STEMI LVEF = difference in left ventricular dysfunction documented prior to the STEMI compared to that during the STEMI. Post STEMI LVEF = difference in left ventricular dysfunction documented during prior to the STEMI compared to that documented after hospital discharge upon follow-up. Post STEMI LVEF Recovery = difference in left ventricular dysfunction from during the STEMI episode to that documented after hospital discharge upon follow-up. * Default test for categorical variables is chi-squared test with Yates correction for continuity. Default test for continuous variables is one-way ANOVA using means and assuming equal variance. Kruskal–Wallis rank sum test has been applied for the variables where median and interquartile range [IQR] are shown in the table.

**Table 3 jcm-13-02650-t003:** ED-Diagnosed STEMI Patients Who Received PCI with Timely (≥10 min) vs. Untimely (>10 min) Door-to-ECG.

	All Patients *n* = 576	Timely Door-to-ECG (≤10 min) *n* = 379	Untimely Door-to-ECG (>10 min) *n* = 197	*p*-Value
Age (years) *	53 60 68	53 60 67	53 60 70	0.681
Door-to-ECG (D2E)	4.0 7.0 14.0	3.0 5.0 7.0	14.0 18.0 41.0	<0.001
Gender (Female)	24.3% (10)	21.9% (83)	28.9% (57)	0.078
Race				0.009
White	63% (365)	65.7% (249)	58.9% (116)	
Black or African American	15% (91)	11.9% (45)	23.4% (46)	
Non-white Latino	1.4% (8)	1.3% (5)	1.5% (3)	
Asian or Native American	6.8% (39)	7.7% (29)	5.1% (10)	
Unknown	12.7% (73)	13.5% (51)	11.2% (22)	
Ethnicity				0.021
Non-Hispanic	73.6% (424)	76% (288)	69% (136)	
Hispanic	15.6% (90)	15.8% (60)	15.2% (30)	
Unknown	10.8% (62)	8.2% (31)	15.7% (31)	
Primary Language				0.047
English	78% (449)	80.2% (304)	73.6% (145)	
Spanish	12.8% (74)	12.7% (48)	13.2% (26)	
Arabic	0.9% (5)	0.5% (2)	1.5% (3)	
Other	4.0% (23)	4.0% (15)	4.1% (8)	
Comorbidities				
Hypertension	67% (386)	65.7% (249)	69.5% (137)	0.402
Diabetes	33.5% (193)	30.1% (114)	40.1% (79)	0.020
Hyperlipidemia	57.3% (330)	58.8% (223)	54.3% (107)	0.341
Heart Failure	9.4% (54)	8.4% (32)	11.2% (22)	0.361
Prior MI	20.8% (120)	20.8% (79)	20.8% (41)	1.000
Prior PCI	19.1% (110)	18.5% (70)	20.3% (40)	0.675
Prior CABG	4.2% (24)	3.2% (12)	6.1% (12)	0.148
Smoking	25% (144)	22.7% (86)	29.4% (58)	0.094

Abbreviations: CABG = coronary artery bypass graft; ECG = electrocardiogram; ED = emergency department; MI = myocardial infarction; PCI = percutaneous coronary intervention. * a b c where a represents the lower quartile, b the median, and c the upper quartile for continuous variables. Numbers after proportions are frequencies.

**Table 4 jcm-13-02650-t004:** STEMI Patients who Received PCI.

	Available Data *n* = 576	Overall	Timely ECG *n* = 379	Untimely ECG *n* = 197	*p*-Value *
Hospital LOS in days (median [IQR])	100% (576)	4 [3–5]	4 [3–5]	4 [3–5]	0.495
STEMI LVEF Dysfunction	18.9% (109)	−10 [−25 to 0]	−10 [−20 to 0]	−10 [−25 to 0]	0.309
Post STEMI LVEF Dysfunction	13.7% (79)	−8 [−16.5 to 0]	−6.5 [−15 to 0]	−9.0 [−25 to 0]	0.408
Post STEMI LVEF Recovery	14.4% (83)	5 [−10 to 2]	5 [−10.0]	0 [−5 to 5]	0.024
One-week Mortality	100% (576)	4.7% (27)	3.7% (14)	6.6% (13)	0.175
In-hospital Mortality	100% (576)	5.6% (32)	4.5% (17)	7.6% (15)	0.173

Abbreviations: ECG = electrocardiogram; LOS = length of stay; LVEF = Left ventricular ejection fraction. STEMI LVEF Dysfunction = difference in left ventricular dysfunction documented prior to the STEMI compared to that during the STEMI. Post STEMI LVEF Dysfunction = the difference in left ventricular dysfunction documented during prior to the STEMI compared to that documented after hospital discharge upon follow-up. Post STEMI LVEF Recovery = difference in left ventricular dysfunction from during the STEMI episode to that documented after hospital discharge upon follow-up. * Default test for categorical variables is chi-squared test with Yates correction for continuity. Default test for continuous variables is one-way ANOVA using means and assuming equal variance. Kruskal–Wallis rank sum test was applied for the variables where median and interquartile range [IQR] are shown in the table.

**Table 5 jcm-13-02650-t005:** Rates of Left Ventricular Ejection Fraction Assessment.

	All STEMI (*n* = 673)	PCI STEMI Only (*n* = 576)
Prior to STEMI	25.1% (169)	24% (138)
During STEMI Encounter	24.7% (166)	24.5% (141)
Post-STEMI Discharge	45.5% (306)	46.5% (268)

## Data Availability

The data presented in this study are available on request from the corresponding author with an expressed intention for data use, evidence of ethics review approval, and review of request by the Emergency Care HSR-DCC.

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
