# Peer review of "Shorter Door-to-ECG Time Is Associated with Improved Mortality in STEMI Patients"

_jcm, 2024, doi:10.3390/jcm13092650_

Round 1

Reviewer 1 Report (Previous Reviewer 1)

Comments and Suggestions for Authors

The paper is interesting and the authors have revised the comments accordingly.

Author Response

Thank you for your favorable review and comment.

Reviewer 2 Report (Previous Reviewer 2)

Comments and Suggestions for Authors

1.       Pain/symptoms-to-ECG time is a critical factor that can affect the results, it is necessary to consider it as a variable.

2.       Comprehensive patient demographics in this study are important for interpreting the results.

3.       Both LVEF values and changes in LVEF should be considered in the assessment.

4.       The manuscript could further discuss the practical implications for clinical practice, e.g., how to improve LVEF procedures/documentation, the generalizability of this study results.

Author Response

Pain/symptoms-to-ECG time is a critical factor that can affect the results, it is necessary to consider it as a variable.

Response: Thank you for sharing this perspective, however, this is outside the scope of this investigation

Comprehensive patient demographics in this study are important for interpreting the results

Response: As this study was examining trends across several US based hospitals, and the demographics for his cohort have been previously published, we did not feel that more extensive demographics contributed to the novelty of his paper. The results are included in aggregate, without demographic stratification apart from timely vs untimely ECGs. 

Both LVEF values and changes in LVEF should be considered in the assessment.

Response: We found changes in LVEF to be more reflective of the impact of the acute ischemic event.

The manuscript could further discuss the practical implications for clinical practice, e.g., how to improve LVEF procedures/documentation, the generalizability of this study results.

ResponseWe considered including this, however, this is well covered in existing practice guideline which makes the lack of occurance even more concerning.

Reviewer 3 Report (Previous Reviewer 3)

Comments and Suggestions for Authors

The authors made a number of recommended changes. However, in the list of references, half of the sources are older than 5 years ago; it is recommended to correct this.

Author Response

The authors made a number of recommended changes. However, in the list of references, half of the sources are older than 5 years ago; it is recommended to correct this.

Much strong work in the area of STEMI care has been established for many decades. In addition, guidelines currently in effect are cited of which many are > 5 years old. Omitting them would undermine the message that the recommendations we examine are not new, yet there is still a marked clinical practice gap with clinical consequences that should not be ignored.

Thank you for the opportunity to respond.

This manuscript is a resubmission of an earlier submission. The following is a list of the peer review reports and author responses from that submission.

Round 1

Reviewer 1 Report

Comments and Suggestions for Authors

The manuscript contains interesting data and conclusions, but a few comments deserve consideration:

In the section Results: Table 3 is incomplete: there is no data about LVEF, mentioned in the text. There is no data about length of stay; all of these are mentioned in Abbreviations. The data must be shown. Discussion: The following: “the in-hospital mortality of 4.6% in a comparable US-based cohort including all STEMI patients (Table 3).” But, the data about mortality are not shown in Table 3.

Author Response

Reviewer 1 Comment 1: The manuscript contains interesting data and conclusions, but a few comments deserve consideration:

Response: Thank you.  We have addressed your concerns below. 

Reviewer 1 Comment 2: In the section Results: Table 3 is incomplete: there is no data about LVEF, mentioned in the text.

Response:  Thank you for identifying this. LVEF data for the PCI cohort is now included in Table 4 which has been added to the manuscript.

Reviewer 1 Comment 3: There is no data about length of stay; all of these are mentioned in Abbreviations. The data must be shown. 

Response: Hospital LOS is included in Tables 2 and 4. Table 4 was missing and has now been included to support the text.

Reviewer 1 Comment 4: Discussion: The following: “the in-hospital mortality of 4.6% in a comparable US-based cohort including all STEMI patients (Table 3).” But, the data about mortality are not shown in Table 3.

Response: This has been corrected to say “Table 4.” Table 4 was missing but is now included.

Reviewer 2 Report

Comments and Suggestions for Authors

It is well known that timely performing electrocardiography (ECG) is crucial for early detection of STEMI, and reducing door-to-electrocardiography time is an important step in adhering to the recommended door-to-balloon times. The Authors conducted a multicenter study, compared the mortality and LVEF dysfunction between patients who received timely vs untimely ECG. I have the following suggestions/concerns for the authors:

1.       Although among the components of the door-to-balloon process, the door-to-ECG process is the most critical delay interval, the time from the patient’s arrival at hospital to reperfusion is most strongly associated with the morbidity and mortality of patients with STEMI. The authors are also required to provide the door-to-balloon time, and the correlations between door-to-ECG times and door-to-balloon times.

2.       There are studies showing the paintoballoon interval impacted mortality, the paintoballoon times need to be included if applicable. What about pain/ symptoms-to-ECG time?

3.       Patient demographics needs to include more information: symptoms, mode and timing of the ED visit, heart rate, systolic blood pressure, diastolic blood pressure, Killip score, risk factors, medication, portion of MI.

4.       The author mentioned the LVEF dysfunction, then the LVEF values/ Grade of systolic function need to be analyzed.

5.       In addition to grouping according to >10 minutes/10 minutes, it is strongly recommended to further subgrouping of the two groups, e.g., ECG occurring during intake, triage, and main ED care periods. Or further subgroup by time, divide each group into two or three subgroups according to the distribution of data.

6.       Factors associated with delayed diagnosis and/or delayed treatment of STEMI.

7.       Comparison of the door-to-ECG time and mortality among different centers/ geographical strata.

8.       ACC/AHA has recommended the target times of DTE within 10 min and DTB within 90 min, respectively, which have become the benchmark for the management of acute coronary syndrome worldwide. It is well known that “time is myocardium”. What contributions do this study make to emergency practice? Or key/ new clinical implication?

9.       The study size was underpowered, and prospective study is necessary.

Author Response

Reviewer 2 Comment 1:   Although among the components of the door-to-balloon process, the door-to-ECG process is the most critical delay interval, the time from the patient’s arrival at hospital to reperfusion is most strongly associated with the morbidity and mortality of patients with STEMI. The authors are also required to provide the door-to-balloon time, and the correlations between door-to-ECG times and door-to-balloon times.

Response: We cite work on the established relationship between door-to-ECG time and its association between with door-to-balloon time (D2B) with citation 1-10. We have also summarized our prior work on door-to-ECG time quantifying its association with ECG-to-balloon time where we found that every minute of D2E time is associated with 1.24 min of ECG-to-Balloon time. The design of the analysis reported in his paper is specifically focused on the association of timely vs untimely D2E with mortality and HF.

Reviewer 2 Comment 2:  There are studies showing the pain‐to‐balloon interval impacted mortality, the pain‐to‐balloon times need to be included if applicable. What about pain/ symptoms-to-ECG time?

Response: This is indeed the case. There is variable reporting on the validity of symptom (or pain)  to ECG time. This paper used the reference point of the treating ED where door-to-ECG is the metric used o quantify timely diagnostic care.  Examining symptom or pain to ECG time was outside of the scope of this scientific evaluation.

Reviewer 2 Comment 3.       Patient demographics needs to include more information: symptoms, mode and timing of the ED visit, heart rate, systolic blood pressure, diastolic blood pressure, Killip score, risk factors, medication, portion of MI.

Response: We have previously published on mode of arrival and ED arrival time (during or after cath lab business hours) for this cohort.  So including the visit and process characteristics would not be unique contributions to the literature.  We share the references below for the reviewer.  In those papers we explored patient and operational factors potentially influencing door to treatment time.  In this paper we focus on quantifying morbidity and mortality, and include only patient characteristic. This was a choice we made particularly since there were no significant differences in mode of arrival or ED arrival time distribution among those with timely vs untimely care in all ED-diagnosed STEMI patients (n=676) nor the more restricted population who received PCI (n=576). We also have previously published on reported symptoms in this cohort where we found the hierarchy of symptoms reporting among those with timely vs untimely care is nearly identical.  However, the proportion of patients reporting the more classic symptoms of chest pain and shortness of breath is reduced making atypical symptoms disproportionally represented among those with untimely care.

Yiadom MY, Gong W, Patterson BW, Baugh CW, Mills AM, Gavin N, Podolsky SR, Salazar G, Mumma BE, Tanski M, Hadley K, et al. Fallacy of Median Door‐to‐ECG Time: Hidden Opportunities for STEMI Screening Improvement. Journal of the American Heart Association. 2022 May 3;11(9):e024067.

Yiadom MY, Gong W, Bloos S, Liu D. 287 influence of time to diagnosis on time to percutaneous coronary intervention for emergency department ST elevation myocardial infarction (STEMI) patients: door to ECG matters. Annals of Emergency Medicine. 2022 Oct 1;80(4):S125.

Yiadom MY, Gong W, Bloos S, Liu D, et al. Influence of time to diagnosis on time to percutaneous coronary intervention for emergency department ST elevation myocardial infarction (STEMI) patients: door to ECG matters. Journal of the American College of Emergency Physicians. In press

We did not include systolic blood pressure or diastolic blood pressure as this is not routinely available for this ED-diagnosed cohort until well after decision for an ECG is made

Killip score, portion of MI and medication reflect information collected reliably well after the patients encounter with the emergency department and door-to-ecg care interval and thus not found to be pertinent to the care phase studied in this investigation.

Reviewer 2 Comment 4: The author mentioned the LVEF dysfunction, then the LVEF values/ Grade of systolic function need to be analyzed.

Response: Cited guidelines advise for the use of LVEF alone, as a result this data is what was collected from all 10 centers.  We did not grade dysfunction, rather we calculated the change in LVEF as described in the methods and illustrated in Figure 1.

Reviewer 2 Comment 5:  In addition to grouping according to >10 minutes/≤10 minutes, it is strongly recommended to further subgrouping of the two groups, e.g., ECG occurring during intake, triage, and main ED care periods. Or further subgroup by time, divide each group into two or three subgroups according to the distribution of data.

Response: We have previously published on the association of door-to-ECG time with ECG-to-balloon time using this approach.  However, for this analysis examining mortality we did not find this would add to the investigation. We chose to dichotomize D2E at 10 minutes to align with existing guidelines recommending the acquisition of an ECG within 10 minutes and quantifying the consequence of not meeting this metric for each patient with STEMI.

Yiadom MY, Gong W, Patterson BW, Baugh CW, Mills AM, Gavin N, Podolsky SR, Salazar G, Mumma BE, Tanski M, Hadley K. Fallacy of Median Door‐to‐ECG Time: Hidden Opportunities for STEMI Screening Improvement. Journal of the American Heart Association. 2022 May 3;11(9):e024067.

Yiadom MY, Gong W, Bloos S, Liu D. 287 influence of time to diagnosis on time to percutaneous coronary intervention for emergency department ST elevation myocardial infarction (STEMI) patients: door to ECG matters. Annals of Emergency Medicine. 2022 Oct 1;80(4):S125.

Yiadom MY, Gong W, Bloos S, Liu D, et al. Influence of time to diagnosis on time to percutaneous coronary intervention for emergency department ST elevation myocardial infarction (STEMI) patients: door to ECG matters. Journal of the American College of Emergency Physicians. In press

Reviewer 2 Comment 6:  Factors associated with delayed diagnosis and/or delayed treatment of STEMI.

Response: The focus of this paper was to quantify mortality among those with timely vs delayed diagnosis, so reporting on factors for delayed diagnosis and treatment is out of scope for the research question.  However, we have previously published on these topics for the same cohort.

Yiadom MY, Gong W, Bloos S, Liu D. 287 influence of time to diagnosis on time to percutaneous coronary intervention for emergency department ST elevation myocardial infarction (STEMI) patients: door to ECG matters. Annals of Emergency Medicine. 2022 Oct 1;80(4):S125.

Yiadom MY, Gong W, Bloos S, Liu D, et al. Influence of time to diagnosis on time to percutaneous coronary intervention for emergency department ST elevation myocardial infarction (STEMI) patients: door to ECG matters. Journal of the American College of Emergency Physicians. In press

Yiadom MY, Baugh CW, Jenkins CA, Tanski M, Mumma BE, Vogus TJ, Miller KF, Jackson BE, Lehmann CU, Dorner SC, West JL. Outcome Differences Associated with STEMI Diagnostic Delay: Disparities on the Frontlines of STEMI Care. Circulation: Cardiovascular Quality and Outcomes. 2018 Apr;11(suppl_1):A185-.

Bloos SM, Kaur K, Lang K, Gavin N, Mills AM, Baugh CW, Patterson BW, Podolsky SR, Salazar G, Mumma BE, Tanski M. Comparing the Timeliness of Treatment in Younger vs. Older patients with ST-segment elevation myocardial infarction: a multi-center cohort study. The Journal of Emergency Medicine. 2021 Jun 1;60(6):716-28.

Yiadom MY, Baugh CW, Jenkins CA, Tanski M, Mumma BE, Vogus TJ, Miller KF, Jackson BE, Lehmann CU, Dorner SC, West JL. Outcome Differences Associated with STEMI Diagnostic Delay: Disparities on the Frontlines of STEMI Care. Circulation: Cardiovascular Quality and Outcomes. 2018 Apr;11(suppl_1):A185-.

Reviewer 2 Comment 7:   Comparison of the door-to-ECG time and mortality among different centers/ geographical strata.

Response: We did not pursue this analysis given the relative rarity of STEMI and low incidence of mortality.  This necessitated a multi-centered study. So stratifying results by center would include very small numbers that would not provide reliable estimates for a generalizable message.  We focused on quantifying mortality in a more contemporary US cohort as this was a novel contribution to the literature where we had reliable results.

Reviewer 2 Comment 8 ACC/AHA has recommended the target times of DTE within 10 min and DTB within 90 min, respectively, which have become the benchmark for the management of acute coronary syndrome worldwide. It is well known that “time is myocardium”. What contributions do this study make to emergency practice? Or key/ new clinical implication?

Response: Thank you for this question.  This study makes 3 important contributions.

  1. Quantifying mortality in more a contemporary US STEMI patient cohort
  2. Quantifying mortality in an ED-diagnosed cohort, specifically
  3. Quantifying the association of mortality with timely vs untimely care. Its been previously established that faster time to treatment is associated with less LVEF dysfunction and morbidity, but not diagnosis.  So this not only validates that screening and diagnostic emergency care phase, that identified patients who need treatment is valuable but quantifies the mortality and LVEF dysfunction cost of untimely care.  This is included in the Conclusion of the abstract and main paper and the main paper’s Discussion.

Reviewer 2 Comment 9.       The study size was underpowered, and prospective study is necessary.

Response: A larger study sample may have made some of the magnitude differences in LVEF and mortality have a stronger statistical signal.  However, the Emergency STEMI Care Registry is a unique 10 centered registry with 3 years of data, and the findings of our analysis were unique contributions to the literature as discussed above in response to Reviewer 2 Comment 8. We agree that a prospective cohort study is an excellent follow up to this retrospective analysis given our findings.

Reviewer 3 Report

Comments and Suggestions for Authors

Despite the fact that the article is quite interesting, I came up with a number of notes:

1. It is necessary to remove abbreviations from the title.

2. The title does not correspond to the conclusions

3. It is recommended to use abbreviations when repeating words in the text.

4. It is recommended to track the numbering of links in the text, for example, link 13 comes after link 20.

5. Also a question: is it worth assessing EF if it affected less than half of the patients? Why you could not measured it for all patients?

Author Response

Reviewer 3 Comment 1. It is necessary to remove abbreviations from the title.

Response: The title includes the abbreviations STEMI, ECG and ED. These are critical terms for the paper and important to include in the title.  Expanding each term would exceed typical words displayed in searches for pubmed, google scholar, web of science, etc. We balanced this with these terms being familiar to most engaged with cardiovascular disease literature. We are happy to change if the editor feels this is critical

Reviewer 3 Comment 2. The title does not correspond to the conclusions

Response: We found shorter D2E was associated with statistically significant mortality among those receiving PCI. We did not find an association with LVEF dysfunction that was statistically significant, although magnitude differences suggest there is a notable signal that should not be ignored. As a result, we only included our mortality findings in the study title which needs to be brief.  However, we felt it was important to highlight the LVEF measurement limitations that are a population health challenge. These points were the content included in the conclusion of the abstract and main paper.

Reviewer 3 Comment 3. It is recommended to use abbreviations when repeating words in the text.

Response: We repeat the full description of some of our time-to-XXX terms, despite abbreviations being presented at the beginning, to improve readability. This was done reflecting on feedback we have received from other journals.  We have not changed.  We are glad for this to be changed if this is preferred by the journal.

Reviewer 3 Comment 4. It is recommended to track the numbering of links in the text, for example, link 13 comes after link 20.

Response: Citation 13 first appears at the end of the first paragraph which is before citation 20.

Reviewer 3 Comment 5. Also a question: is it worth assessing EF if it affected less than half of the patients? Why you could not measured it for all patients?

Response: This was a retrospective study so the frequency of LVEF assessment was what was observed as it occurred in routine care. As noted in the discussion, LVEF assessment for study patient is recommended by international guidelines to encourage the identification of those with low LVEF. This group has potential for functional and mortality improvement if goal directed medication therapy is initiated early. This opportunity is missed if LVEF is never assessed.  

Round 2

Reviewer 2 Report

Comments and Suggestions for Authors

The authors made some improvements and addressed some comments/concerns.

Author Response

Thank you for recognizing our revisions.  We appreciate you taking the time to review.

Reviewer 3 Report

Comments and Suggestions for Authors

1. The word "ED" could definitely be omitted from the title, which would reduce the number of abbreviations in the title

2. The conclusion and the very essence of the article are extremely questionable, since ECG assessment in patients with STEMI within the first 10 minutes, as well as the need to assess EF in patients with MI are included in existing recommendations for the management of patients with STEMI (https://doi.org/10.1093/eurheartj/ehad191). What practical value does this research have?

3. The authors made the appropriate edits, but only mentioned references No. 13 and 20.

Author Response

Reviewer 3 Comment 1: The word "ED" could definitely be omitted from the title, which would reduce the number of abbreviations in the title

Response: Edited as requested.

Reviewer 3 Comment 2: The conclusion and the very essence of the article are extremely questionable, since ECG assessment in patients with STEMI within the first 10 minutes, as well as the need to assess EF in patients with MI are included in existing recommendations for the management of patients with STEMI (https://doi.org/10.1093/eurheartj/ehad191). What practical value does this research have?

Response: We agree with the reviewer that 1) ECG assessments in the first 10 minute and 2) assessing EFs in patients with MI is included in existing guidelines and supported by prior literature.  This is noted in the recently updated 2023 ESC guidelines they shared, and has been included in prior guidelines in the Americans, Europe and internationally.  The contribution our work makes to science and clinical practice is that despite guidelines:

  1. ECGs are not assessed in 10 minutes in many patients. We directly quantify the clinical impact of guideline non-compliance in an ED-diagnosed cohort.
  2. LVEF is not assessed in all MI patients. We quantify the lack of compliance with guidelines, and note how it limits access to targeted therapies.

The practical value of this research is highlighting the gap between guidelines and practice. We highlight this with our analysis, and summarized the gap in our conclusion.  These are opportunities to improve care delivery to patients. Guidelines are excellent as are the periodic updates that guide best practice, but there is still more work to do to make it typical practice. Research highlighting the gap is critical to create awareness and motivation for improvement.

Reviewer 3 Comment 3: The authors made the appropriate edits, but only mentioned references No. 13 and 20.

Response: Upon re-review we agree that all edits as noted have been made